# Learning to see colours: Biologically relevant virtual staining for adipocyte cell images

**Håkan Wieslander** [1][⊛], **Ankit Gupta** [1][⊛], **Ebba Bergman** [2], **Erik Hallström** [1], **Philip John Harrison** [2]*

**1** Department of Information Technology, Uppsala University, Uppsala, Sweden, **2** Department of Pharmaceutical Biosciences, Uppsala University, Uppsala, Sweden

⊛ These authors contributed equally to this work.
* philip.harrison@farmbio.uu.se

**Data Availability Statement:** A sample of the data is freely available at https://www.ai.se/en/node/81535/adipocyte-cell-imaging-challenge and access to the entire dataset can be requested from AI Sweden (https://www.ai.se/en). All of the code

## Abstract

Fluorescence microscopy, which visualizes cellular components with fluorescent stains, is an invaluable method in image cytometry. From these images various cellular features can be extracted. Together these features form phenotypes that can be used to determine effective drug therapies, such as those based on nanomedicines. Unfortunately, fluorescence microscopy is time-consuming, expensive, labour intensive, and toxic to the cells. Bright-field images lack these downsides but also lack the clear contrast of the cellular components and hence are difficult to use for downstream analysis. Generating the fluorescence images directly from bright-field images using virtual staining (also known as "label-free prediction" and "in-silico labeling") can get the best of both worlds, but can be very challenging to do for poorly visible cellular structures in the bright-field images. To tackle this problem deep learning models were explored to learn the mapping between bright-field and fluorescence images for adipocyte cell images. The models were tailored for each imaging channel, paying particular attention to the various challenges in each case, and those with the highest fidelity in extracted cell-level features were selected. The solutions included utilizing privileged information for the nuclear channel, and using image gradient information and adversarial training for the lipids channel. The former resulted in better morphological and count features and the latter resulted in more faithfully captured defects in the lipids, which are key features required for downstream analysis of these channels.

## 1 Introduction

Nanomedicine uptake and effect on fat cells (adipocytes) can be explored using microscopy imaging techniques applied to stem cell derived cell cultures. The lipid droplets within the adipocytes play a key role in metabolism and are implicated in several pathologies, including cancer, diabetes and obesity [1]. They provide fuel for the organism and supply a safeguard for energy fluctuations [2]. Membrane surfaces of lipid droplets can contain hundreds of different proteins (such as perilipins, enzymes and trafficking proteins) that allow them to function as energy repositories and interact with other cellular components [3]. The metabolic demands

and trained models to accompany this manuscript are available on GitHub (https://github.com/aktgpt/brevis).

**Funding:** This project was financially supported by the Swedish Foundation for Strategic Research (https://strategiska.se/en/, grants BD150008 and ARC19-0016). These grants were awarded to Carolina Wählby (the PhD supervisor of the five authors on this paper).

**Competing interests:** No authors have competing interests.

of nanomedicine transfection can therefore cause physical remodeling of the lipid droplets as the cells respond to the increased energy demands and the resulting phenotypic signatures can be used to evaluate nanomedicine efficacy. Therapies that modify adipocyte phenotypes can also be used to adjust metabolic profiles towards more healthy states by encouraging catabolic lipid processing [4]. Furthermore, using nanoparticle delivery of mRNA to adipocytes, via subcutaneous injection, is a very promising next-generation approach for protein replacement therapies and vaccination [5]. Such mRNA based vaccinations are also currently being developed for Covid-19 [6].

A typical workflow in image cytometry utilizes fluorescent staining and imaging to extract relevant features from cells (such as morphology, count and intensity) and subsequently builds classifiers for various purposes based on these features. Labelling with fluorescent dyes is a well-established and effective method for identifying cellular components. However, imaging fluorescence stains does not come without drawbacks. They require sample preparation, are time-consuming, expensive and labor intensive to develop, and are often toxic to the cells. For example, Hoechst staining is commonly used to visualize cell nuclei because this blue-fluorescent dye binds to DNA, this binding however tends to disrupt normal DNA processing. The fluorescent dyes must also be excited using relatively short high-energy wavelengths resulting in the production of toxic free-radicals. While these problems can be mitigated using experimental design, they are impossible to avoid completely, particularly when performing time-lapse imaging experiments with many exposures [7]. If possible it would be better to circumvent the need for these fluorescent stains and short wavelength light exposures. Additionally, there is a limit to the number of cellular components that can be imaged simultaneously for a given sample due to spectral overlap of the requisite fluorescence dyes.

Bright-field images, on the other hand, are significantly easier to acquire, require very little sample preparation, and are not toxic to the cells. Hence, a less invasive method (and the one proposed herein) would be to predict the fluorescence images directly from their corresponding bright-field images. This translation needs to be as accurate as possible to ensure that the downstream analysis from the generated images is very close to that obtained from the real images.

In recent years deep learning has shown impressive results for image reconstruction and enhancement tasks in image cytometry including deblurring, denoising and super-resolution, and has recently shown promise as a virtual staining method [8, 9]. The "in-silico labeling" approach used by [10] is a particularly noteworthy example of the latter. They proposed a U-Net deep learning architecture [11] and inception-inspired modules [12] to generate fluorescence images given transmitted-light images as input. Their model was trained with pixel-wise cross entropy loss. Using only pixel-wise loss, however, their generated images in some instances lacked global coherence [13]. Also noted, in their virtual staining method for histological images, that training using only pixel-wise loss led to good localization but to poor resolution of finer cellular structures. There are many loss functions and architectural choices that can be explored to combat these issues. For instance, for generating crisp, realistic looking images generative adversarial networks (GANs; [14]) have shown great promise. As opposed to computing the losses over pixels, GANs work with losses at the image-level (or across patches of an image) and push the generated images towards the natural image manifold [15]. In its basic form a GAN plays out a zero-sum game between two networks—a generator and a discriminator. The generator creates counterfeit images which it hopes to deceive the discriminator with, whilst the discriminator attempts to correctly classify the real and fake images. Convergence is reached when the discriminator can no longer tell the difference between the images. For image-to-image translation tasks conditional GANs are often applied. The conditional discriminator sees both the input to the generator and the fake or real outputs. This can help alleviate the problem of artifacts that GANs can produce and was used in, for instance,

the Pix2Pix algorithm [16]. In the field of virtual staining, conditional GANs have been used by, for example, [17] for histological staining of prostate core RGB images, and by [18] for histological staining of skin, liver and kidney tissue sections. Additional virtual staining methods and applications of note include the 3D U-Net-based approach used in [19], for 3D fluorescence reconstruction, and the dual adversarially trained autoencoder model used in [20]. This latter method used nuclear and cell fluorescence stains to train a first autoencoder, learning a probabilistic encoding for cell and nuclear shape, upon which a second "wedded" autoencoder was used for generating additional subcellular structures, conditional on this encoding.

A relatively new paradigm within machine learning is Learning Using Privileged Information (LUPI; [21]). The key idea of the LUPI paradigm is that the machine learner, the "student", has an "intelligent teacher" that provides additional (privileged) information during the training phase, to improve and accelerate the learning process. Through LUPI the learner gains a better concept of similarity between training objects and also receives hidden insights (explanations from the teacher) to guide decision rule formation. The privileged information can be anything that potentially aids the networks during the training process, but that is not required or available during inference. The privileged information can help to focus the network's attention on locations in the data that are difficult to predict. The way in which this is done in deep learning is to develop separate network streams, one for the main task and one based on the privileged information, and then to incorporate some means of guiding the main model with insights from the privileged information stream [22–26].

In bright-field images the nuclei are often not visually apparent and hence generating their fluorescence counterpart can be challenging. If there are significant errors in these generated fluorescence images the downstream segmentation of the nuclei will fail, a step that is crucial for obtaining subsequent measures of nuclear features. Furthermore a common approach for downstream analysis of the images is to use a CellProfiler [27] pipeline in which the nuclei (from the nuclear stain) are used for providing seeds to the segmentation algorithms that delineate the cell boundaries (the cytoplasm). Hence, the nuclear channel is very important and needs careful consideration. As a potential aid to the nuclear model we therefore apply LUPI using segmentation masks as privileged information. To the best of our knowledge, the work in this paper presents the first application of LUPI in virtual staining for image cytometry.

A shortcoming of several virtual staining studies in cytometry is an overly strong comparison of the true and generated images at the image level. For biological relevance the generated images should be compared based on various features extracted from the different cellular components. Evaluating synthetic images based on derived features has been proposed previously in [28]. In this paper, we propose a method to generate fluorescent labels for adipocyte cell images directly from bright-field z-stacks. This is done by constructing three different models, one each for nuclei, cytoplasm, and lipid droplets. The faithfulness of this translation is compared at both the image and feature level, but with more emphasis on the features.

The modelling and results presented in this paper were based on our winning solution to the *Adipocyte Cell Imaging* challenge hosted by AstraZeneca AB (AZ) and AI Sweden during November 2020. The features evaluated in the CellProfiler pipelines were based on those used to judge the competition. An overview of our proposed method and workflow is shown in Fig 1. A GitHub repository providing the code base for our modelling solutions is available at https://github.com/aktgpt/brevis.

## 2 Data and evaluation

The data under consideration consists of adiopocyte cells imaged at 60x magnification. The cells were stained with Hoechst 33342 (blue, for the nuclei), Bodipy (green, for the lipid

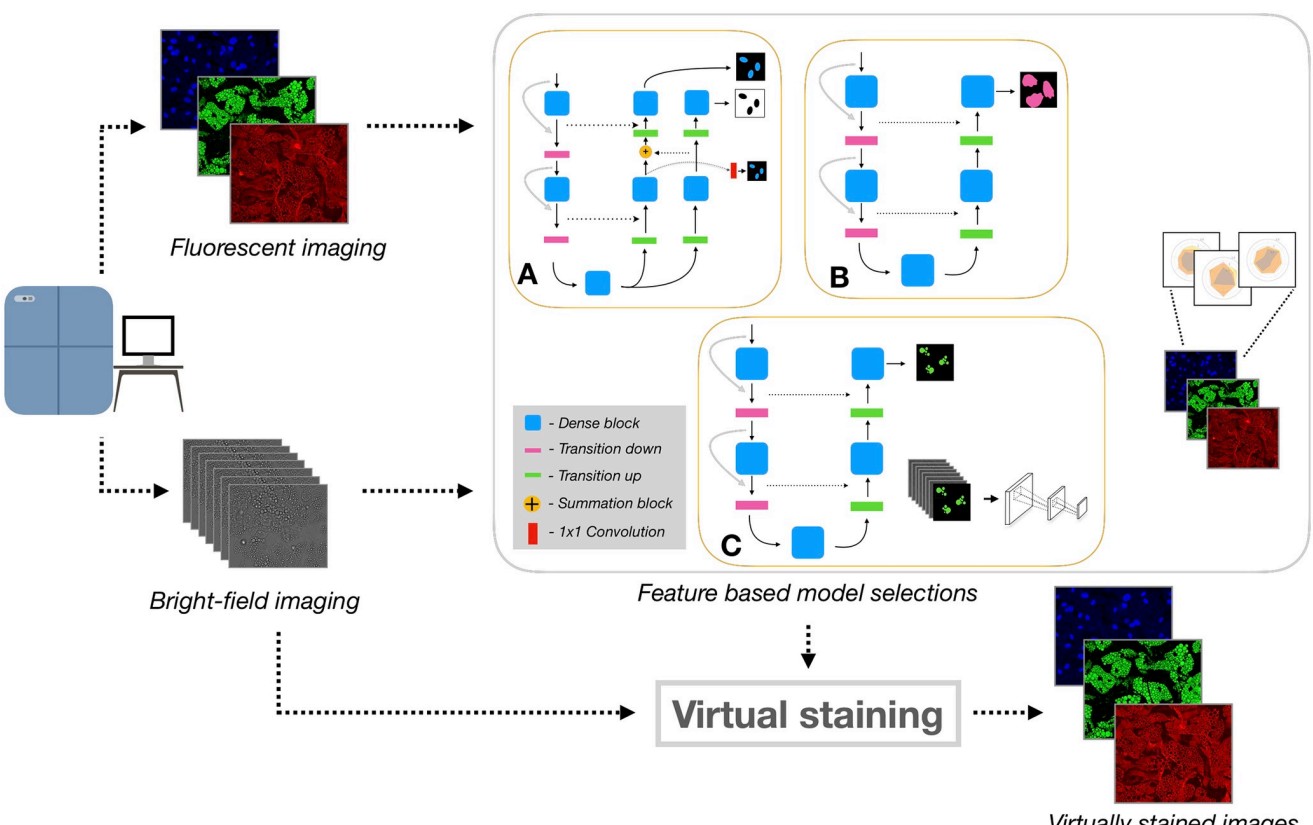

*Fluorescent imaging*

*Bright-field imaging*

- *Dense block*
- *Transition down*
- *Transition up*
- *Summation block*
- *1x1 Convolution*

*Feature based model selections*

**Virtual staining**

*Virtually stained images*

**Fig 1. Proposed method and workflow.** Cell cultures are imaged by fluorescence and bright-field imaging. These images are used to train and compare specialized models for generating each stain, based on biologically relevant features. The selected models are then used to virtually stain bright-field images. A. Nuclear virtual staining model which simultaneously learns the nuclear segmentation (the privileged information model, the outer-most decoder path) and the generated image for the nuclear channel (the inner-most decoder path), B. Virtual staining model for the cytoplasm and C. Virtual staining model for lipid droplets, the model utilizes a patch-level discriminator, used for adversarial training.

droplets) and Cell Tracker Deep Red (red, for the cytoplasm) and imaged with a robotic confocal microscope (Yokogawa CV7000). The cells were imaged with a bright-field z-stack of 7 images with an exposure of 150 *ms*, the fluorescence with an exposure of 300 *ms* for blue, 300 *ms* for green, and 500 *ms* for red with a numerical aperture of the objective of 1.2. The cells were cultured in 8 wells and imaged at 12 different sites. Each image is approximately 2556 × 2156 pixels with a pixel scale of 0.1083 *μm* and used 16 bits to represent the pixel values. The final dataset (8 wells) consisted of 96 bright-field images, all with corresponding fluorescence images (nuclei, lipid droplets and cytoplasm). The stage of the microscope was kept fixed in the x-y direction during imaging to preserve registration between the bright-field and fluorescence images.

## 2.1 Data preprocessing

The dynamic range of microscopy images are rarely the entire bit-depth (16 bits). The images were therefore scaled to 0-1 using their dynamic range. Segmentation masks for the nuclei were produced with CellProfiler (see section 2.3). For each identified nucleus the center coordinate was extracted to make it possible to crop patches in the original images containing nuclei.

The data was divided into three different splits, each having 5 wells for training and 2 for validation. Excluded from these three splits was one well that was set aside for final evaluation.

## 2.2 Image based evaluation

The generated images were compared to the ground truth images based on the mean absolute error (MAE), normalized towards the median of the ground truth images, and the structural similarity index measure (SSIM) [29]. The MAE measures the absolute deviation in pixel intensities between the generated image and the ground truth, for which a lower value is better. SSIM measures the similarity between overlapping sub-windows in the images based on their means, variances and co-variances and returns a value between zero and one, whereby more similar images have a higher value.

## 2.3 CellProfiler evaluation

From the generated and ground truth images a variety of morphological, intensity and count features were extracted using a CellProfiler pipeline ([27] version 4.06). The different cellular structures were initially identified based on intensity, size and shape, prior to feature extraction. Potential nuclear regions were identified by an adaptive thresholding method. Since the lipids can occlude and alter the appearance of the nuclei a wide range of diameters and shapes were used to filter out irrelevant objects. The objects were then filtered based on shape features to remove incorrectly identified nuclear objects. For the lipid droplets, small and large droplets were identified separately. Since the droplets have similar intensities they are identified by a global threshold across the expected range of diameters for the two groups. Lipids that were clumped together were distinguished based on their intensities. The defects in the lipid droplets were found by morphological reconstruction (a rolling-ball algorithm) where the image is successively eroded and reconstructed to identify dark holes within bright regions. Cytoplasmic regions were identified starting with nuclear objects as seeds. Here the combination of the cytoplasmic images and the lipids (which are contained within the cytoplasm) were utilized by averaging them together. Cell bodies were then identified by an adaptive threshold with a window size larger than that of nuclei. For all cellular structures features were extracted separately and are shown in Table 1.

*Compactness* is a measure of irregularities and holes in the object, where a filled circle will have a score of 1 and more irregular objects will have a score grater than 1. *Form Factor* measures the circularity of objects. *Mean Radius* measures the mean distance of any pixel in the object to a pixel outside of the object. *Solidity* measures the proportion of the area of the object

**Table 1. Extracted features from CellProfiler.**

|  | Nuclei | Cytoplasm | Lipids |
|---|---|---|---|
| Morphology | Area<br>Compactness<br>Form Factor<br>Mean Radius<br>Perimeter<br>Solidity | Area<br>Compactness<br>Form Factor<br>Mean Radius<br>Perimeter<br>Solidity | Area<br>Compactness<br>Form Factor<br>Mean Radius<br>Perimeter<br>Solidity<br>Granularity |
| Intensity | Integrated Intensity<br>Mean Intensity<br>Std Intensity | Integrated Intensity<br>Mean Intensity<br>Std Intensity | Integrated Intensity<br>Mean Intensity<br>Std Intensity |
| Count | Count Nuclei | Count Cells | Count Defective Lipids<br>Count Lipids |

against the convex hull. *Granularity* is a measure of size distribution over a fixed set of scales and gives one value per scale. *Integrated Intensity* is the sum of the pixel intensities within an object. *Mean Intensity* is the average pixel intensity within an object. *Std Intensity* is the standard deviation of the pixel intensities within an object.

The value for each feature is computed as the average over the objects in the image. The features were then compared to the features extracted from the ground truth by the MAE and Spearman correlation (denoted by $\rho$). The absolute error in each case was calculated with respect to the median values of the ground truth, expressed as

$$MAE_{median} = \frac{1}{n} \sum_{i=1}^{n} \frac{(y - \hat{y})}{\tilde{y}} \tag{1}$$

where $\tilde{y}$ is the median value of the target. The correlation is calculated as the the average value across the three splits, calculated by a Fisher transform. The score of each feature group (morphology, count and intensity) is then calculated as the mean of the $MAE_{median}$ of the features in each group.

## 3 Model descriptions

### 3.1 Base model

The foundation of the proposed models is a dense U-Net [30]. Like the traditional U-Net [11] this model consist of a contracting/downsampling path (encoder) and an expanding/upsampling (decoder) path, with concatenating skip connections between the two. These skip connections aid the upsampling path in recovering finer grained spatial information from the downsampling path. The dense U-Net extends this basic U-Net architecture through the use of dense blocks consisting of batch-normalization—ReLU—convolution—dropout operations that are densely connected (within a block) to one another in a feed forward fashion (i.e. there is a skip connection with the output of each quartet of operations to every subsequent quartet). On the contracting path there are also skip connections that hop over each dense block. These dense blocks and additional skip connections result in a network that has multi-scale deep supervision with feature propagation and reuse, permitting the training of deeper networks but with fewer parameters than would otherwise be required (see Fig 2 which illustrates the dense U-Net model). The transition down layers consist of spatial max pooling and the transition up layers pixel-shuffle followed by blurring to reduce checkerboard artifacts in the network outputs, as proposed in [31]. The dense U-Net has shown promising results in, for instance, predicting cell traction forces [32] and image restoration [33].

### 3.2 Nuclei model

Since generation of nuclear fluorescence stains based on bright-field images is a difficult task a LUPI-based framework was utilized. This was achieved through adding a supplementary decoder to the base dense U-Net model (see Fig 3). The additional decoder focuses on predicting the segmentation of nuclei and this privileged information is propagated to the image decoder by pixel-wise summation of the intermediate blocks.

Furthermore, inspired by [22] the network employs a multi-scale loss weight module (LWM, see Fig 4) to assign larger weight to the hard pixels. Each decoder block is passed through a convolutional layer to output the nuclear image predictions at each scale. The losses for each intermediate block were then fused with the segmentation weight mask (SWM) to form a weighted loss for the nuclear image. The SWM is employed to assign more weight to the regions where the segmentation is incorrect.

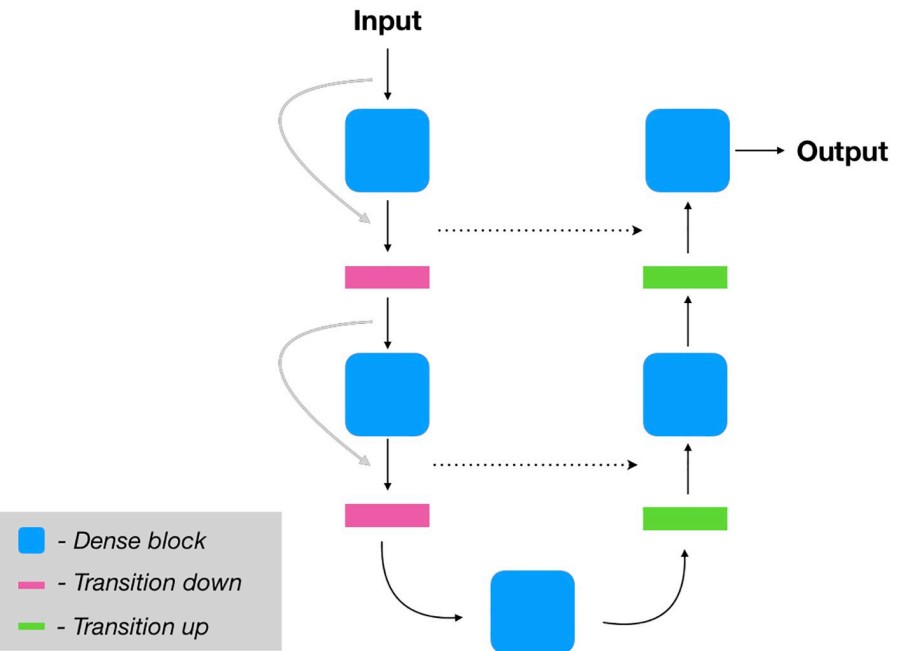

**Fig 2. Illustration of the base model (Dense UNet).** The dense blocks is a series of batch-normalization—ReLU—convolution—dropout that are densely connected. The transition down layer is a spatial max pooling layer and the transition up is a pixel-shuffle followed by a blurring.

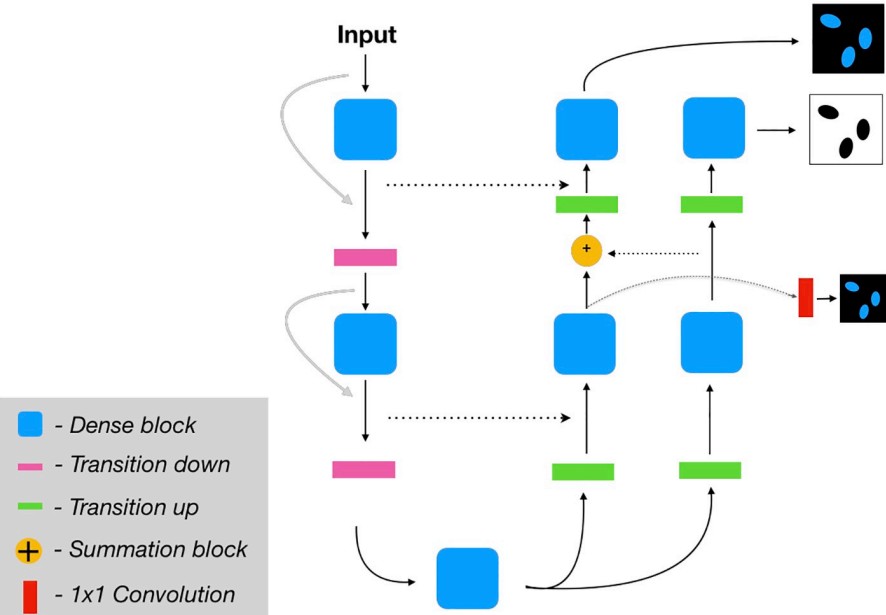

**Fig 3. Proposed network for the nuclear virtual staining model which simultaneously learns the nuclear segmentation (the privileged information model, the outer-most decoder path) and the generated image for the nuclear channel (the inner-most decoder path).** See Fig 4 for more information on how the information from the segmentation decoder is propagated to the image generation decoder.

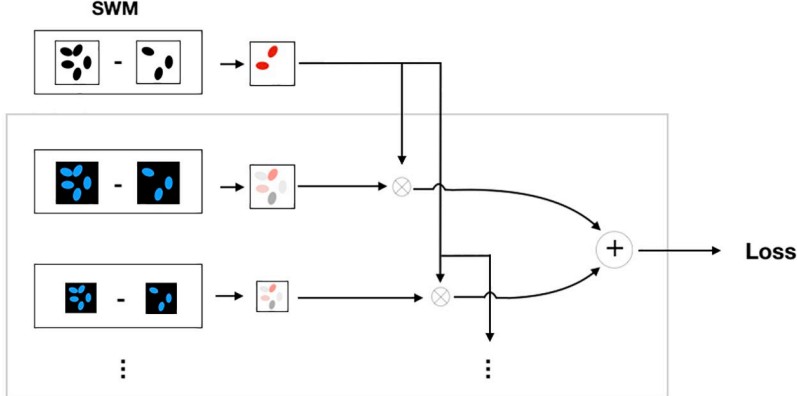

**Fig 4. Illustration of proposed loss weight module (LWM) and segmentation weight mask (SWM).** Each output from intermediate blocks are passed through a 1×1 convolution and compared to the resized ground truth. The loss map is then fused with the SWM to get the final loss.

The model maps the input bright-field image stack $I^{bf}$ to $y^{reg}$ and $y^{seg}$, the regression (based on the nuclear image) and the segmentation respectively. For the segmentation decoder binary cross-entropy loss $\mathcal{L}_{BCE}$ was used. The SWM $Z$ is defined by $Y^{seg}$ (output of the segmentation decoder) as

$$Z_{i,j} = \begin{cases} \eta, & \text{if } Y_{i,j}^{seg} \text{ is misclassified} \\ 1, & \text{otherwise} \end{cases} \tag{2}$$

where $\eta$ is the weight assigned to misclassified pixels. The weighted reconstruction/regression loss $\mathcal{L}_1^{reg}$ is thus defined as

$$\mathcal{L}_{reg}^1 = \sum_{j=1}^{W}\sum_{i=1}^{H} Z_{i,j} \cdot \left\| Y_{i,j}^{reg} - y_{i,j}^{reg} \right\|_1 \tag{3}$$

Similarly, the weighted loss for the outputs of intermediate blocks and resized ground truth is defined as

$$\mathcal{L}_{reg}^{LWM} = \sum_{k=1}^{N} \mathcal{L}_{reg}^{1,k} \tag{4}$$

where $k = \{1, \ldots, N\}$ are the intermediate blocks in the decoder. The final loss for the nuclei channel is then

$$\mathcal{L}_{tot}^{nuclei} = \mathcal{L}_{reg}^1 + \alpha_{LWM} \cdot \mathcal{L}_{reg}^{LWM} + \alpha_{seg} \cdot \mathcal{L}_{BCE} \tag{5}$$

where $\alpha$ is the weight for each loss function.

## 3.3 Lipid model

Virtual staining for the lipids channel was based on the same model as described in section 3.1. To enhance the performance of this model it was trained with an adversarial training scheme, using a conditional GAN approach, where the input to the discriminator included both the bright-field z-stacks and the fake or real fluorescence images. Hence, the discriminator not only asks whether the generated images look real, but also whether or not they look real

conditioned on the bright-field images. Our discriminator also worked at the patch level, rather than at the entire image level, achieved by limiting the receptive field. Working with such patches promotes sharper outputs, better capturing local style characteristics, and to some extent introduces a form of style/texture loss [16]. In combination with this, the models were trained with a Relativistic Average Least Squares GAN (raLSGAN) loss, which is a combination of the relativistic GAN [34] and the least square GAN [35]. In traditional GANs the discriminator estimates the probability that an image is real, whereas in relativistic GANs the discriminator estimates the probability that a real image is more realistic than fake images and conversely, that a fake image is less realistic than a real image. This relativism passes more information to the generator and can significantly improve both generated image quality and training stability. The least square GAN replaces the probability with a least squares loss, i.e. instead of estimating the probability of an image being real or fake it minimizes the least square distance between the two. The losses for the generator and discriminator can be expressed as,

$$\mathcal{L}_G^{adv} = \mathbb{E}[(D(y) - \mathbb{E}[D(G_\theta(x))] + 1)^2]$$
$$+ \; \mathbb{E}[(D(G_\theta(x)) - \mathbb{E}[D(y)] - 1)^2] \tag{6}$$

$$\mathcal{L}_D^{adv} = \mathbb{E}[(D(y) - \mathbb{E}[D(G_(x))] - 1)^2]$$
$$+ \; \mathbb{E}[(D(G_(x)) - \mathbb{E}[D(y)] + 1)^2] \tag{7}$$

Where $D$ is the discriminator and $G$ is the generator.

To complement the adversarial training a mean absolute error loss, $\mathcal{L}^1$, was employed to minimize deviations from the ground truth. To better capture defects in the lipids, a loss function based on (unnormalized) horizontal and vertical image gradients was added. This loss puts extra emphasis into reconstructing edges in the image and is calculated as

$$\mathcal{L}_{grad} = \sum_{j=1}^{W} \sum_{i=1}^{H} \|\nabla x_{i,j} - \nabla y_{i,j}\|_1 \tag{8}$$

where $x$ in the generated image and $y$ is the ground truth image. The total loss for the lipid models is then

$$\mathcal{L}_{tot}^{lipids} = \alpha_{L_1} \cdot \mathcal{L}^1 + \alpha_{grad} \cdot \mathcal{L}_{grad} + \alpha_{adv} \cdot \mathcal{L}_{adv} \tag{9}$$

where $\alpha$ is the weight for each respective loss function. See Fig 5 for an illustration of proposed model used for generating the lipid stain.

### 3.4 Cytoplasm model

The cytoplasm model employed the base model described in section 3.1. The red cytoplasmic stain was quite weak and had high intensity variations. Due to these issues, applying adversarial and/or gradient loss did not show significant improvements over the base model and in fact led to artifacts (bright intensity spots for the adversarial training and checkerboard patterns in some locations for the gradient loss). Hence, this model was trained using only $\mathcal{L}_1$ loss.

### 3.5 Pyramidal weighted inference

To handle inference on large images (in this case 2156×2556 pixels) tiling can be applied, whereby the tiles are inferred by the network and then stitched together to obtain the full

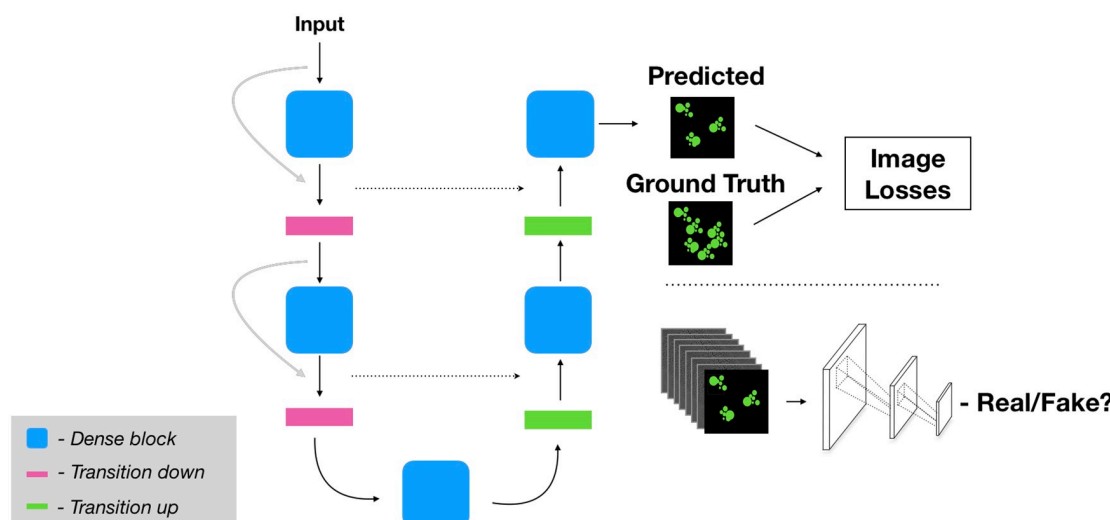

**Fig 5. Proposed model for virtual staining of the lipid droplets.** The base model is extended with a patch-level discriminator, used for adversarial training, in combination with intensity and gradient-based loss functions.

image. We adapt the pyramidal weighted tiling scheme from [36] and create weight masks for each tile, giving a higher weight to central pixels than to pixels at the image boundaries. The weights are reduced in a pyramidal fashion from the central pixel towards the edges. The inference is then performed on overlapping tiles to reduce GPU memory consumption and to have similar receptive fields during training and inference. The outputs are multiplied by the weight mask prior to averaging to create the full image. The method was applied with a tile size of 512×512 pixels and a stride of 256×256 pixels.

## 4 Experiments and results

### 4.1 Model training

For the nuclei model training the loss weights were set to $\alpha_{LWM} = 0.1$, $\alpha_{seg} = 0.05$ and $\eta$ for the SWM was set to 2. The learning rate was set to $10^{-3}$ with batch size of 12. Images were cropped with a size of 512x512 pixels, with a probability of $p = 0.8$ to be cropped around the center of the nuclei (with random offset) and $p = 0.2$ to be cropped at a random location. The maximum offset was set to 256 pixels to ensure that at least one nucleus was present in the image. Augmentations were done with horizontal flipping, vertical flipping, and randomly rotating the images with maximum 180 degrees. The models were then trained for 51,400 iterations.

The weights for the losses for the lipid model were set to $\alpha_{grad} = 5$, $\alpha_{adv} = 0.01$, and $\alpha_{L_1} = 1$. The learning rate for the generator was set to $10^{-4}$ and $10^{-5}$ for the discriminator. The models were trained with a batch size of 4 and augmentations were performed with random crops (of 512x512 pixels), random horizontal and vertical flipping and random 90 degrees rotations. The models were trained for 30,000 iterations. For the first 1500 iterations the generator was trained in isolation to put it in a stronger starting position before introducing the discriminator.

The model for cytoplasm virtual staining took longer to converge and was thus trained for 75,000 iterations. The model was trained with a learning rate of $10^{-4}$ and a batch size of 4, together with the same augmentations as applied for the lipid model.

All the models were trained with AdamW [37] as the optimizer with weight decay of 0.01. The models for the lipids and cytoplasm channels had four dense blocks each in both the encoder and decoder paths. The models for nuclear channel had five blocks in each path, as preliminary experiments showed a higher capacity was beneficial in this case. The dropout rate was set to 0.2 and the growth rate of the dense blocks was set to 12. The best models were saved on the basis of the loss evaluated on the validation set.

## 4.2 Ablation analysis

To explore the effect of each loss and model configuration on the generated images an ablation study was performed. These effects were evaluated based on the pixel-wise mean absolute error and on the three groups of CellProfiler features (morphology, intensity and count). The scores were calculated for the outputs of the models across the three training-validation splits of the data, and represents the mean absolute error between the generated and ground truth images and the mean absolute error for each group of features. For each loss/model configuration a one-sided Mann-Whitney rank test [38] was performed to evaluate the significance of the improvements.

For the nuclear channel, the effect of three model configurations were evaluated: $\mathcal{L}_1$ which uses only the base model described in section 3.1 and MAE as loss function; $\mathcal{L}_1 + \mathcal{L}_{BCE}$, which extends the base model by adding another decoder as described in section 3.2 with only MAE and cross-entropy loss as loss functions, and lastly: $\mathcal{L}_1 + \mathcal{L}_{BCE} + \mathcal{L}_{LWM}$, which also adds the multi-scale loss modules.

For the lipid droplets the addition of gradient loss and adversarial loss was evaluated against using only the L1 loss.

The effect of the addition of each model/loss configuration for both nuclei and lipids can be seen in Table 2.

For the nuclear channel, a performance improvement was seen in count and morphology features with the LWM (see Table 2). These improvements were significant against the base model ($p < 0.05$ for count; $p \ll 0.001$ for morphology). The LWM was also significantly better for morphology when compared against the LUPI-based model without the LWM ($p \ll 0.001$). The LWM focuses the model towards better shape outlines and punishes it for mislocation of nuclei (see Fig 6). This could be due to the fact that the LWM punishes errors in segmentation by increasing the weights of the reconstruction loss in those regions, hence the loss in correctly classified regions gets less attention as the network becomes better at segmentation.

As can be seen in Table 2, for the lipid droplets, an overall performance improvement was gained at the feature level with the inclusion of each additional loss. The gradient loss gave a significant improvement in morphology ($p < 0.05$), whereas for intensity and count the

**Table 2. Effect of each loss/model configuration on the generated images evaluated on the three groups of CellProfiler features for the nuclear and lipids channels.** The scores are the combined mean and standard deviation over the generated images against the ground truth across the three splits.

| Channel | Configuration | MAE | $CP_{Morphology}$ | $CP_{Intensity}$ | $CP_{Count}$ | $CP_{Tot}$ |
|---|---|---|---|---|---|---|
| Nuclei | $\mathcal{L}_1$ | 0.51 ± 0.070 | 0.05 ± 0.023 | 0.10 ± 0.038 | 0.11 ± 0.072 | 0.07 ± 0.022 |
| | $\mathcal{L}_1 + \mathcal{L}_{BCE}$ | 0.46 ± 0.048 | 0.05 ± 0.018 | 0.11 ± 0.051 | 0.08 ± 0.058 | 0.07 ± 0.021 |
| | $\mathcal{L}_1 + \mathcal{L}_{BCE} + \mathcal{L}_{LWM}$ | 0.49 ± 0.049 | 0.03 ± 0.017 | 0.14 ± 0.070 | 0.07 ± 0.057 | 0.07 ± 0.024 |
| Lipid droplets | $\mathcal{L}_1$ | 0.14 ± 0.030 | 0.19 ± 0.028 | 0.42 ± 0.207 | 0.49 ± 0.189 | 0.26 ± 0.057 |
| | $\mathcal{L}_1 + \mathcal{L}_{grad}$ | 0.14 ± 0.030 | 0.18 ± 0.038 | 0.38 ± 0.213 | 0.45 ± 0.184 | 0.24 ± 0.059 |
| | $\mathcal{L}_1 + \mathcal{L}_{grad} + \mathcal{L}_{adv}$ | 0.14 ± 0.029 | 0.16 ± 0.038 | 0.27 ± 0.158 | 0.25 ± 0.162 | 0.19 ± 0.047 |

**Fig 6. A comparison between the LWM based model (applying LUPI) and the base model for virtual staining of the nuclear channel.** (A) Ground truth fluorescence image, (B) Output from LWM model (C) Output from the base model. The arrows in (C) point out a missing nucleus and an incorrectly shaped nucleus for the base model's prediction. Images are displayed with the same dynamic range for visualization purpose.

improvements were not significant (p = 0.122 and p = 0.135 respectively). The largest overall improvement came with the combination of both gradient and adversarial loss (p ≪ 0.001 for all feature groups), where the largest change for the better was for the count features. Both the adversarial and the gradient loss assist in the delineation of the lipids in the reconstruction of the defects within some of the lipids, relative to the base model, as can be seen in Fig 7.

An observation from the ablation studies, for both the nuclear and lipids channels, was that the pixel-wise MAE did not capture the full picture of model performance. For the lipid droplets, training without the adversarial loss gave the best MAE score. As this metric only summarises the closeness between the generated and ground truth images, it cannot capture all the aspects of the image reconstruction fidelity that may be of interest. This metric is also based on both the foreground and background in the images. In many cases the background is of little or no interest. On the other hand, the feature based metrics we explored are based solely upon on the foreground/objects of interest. The feature set was based upon the cell-level features that biologists routinely extract from such data, and hence they provide much better statistics with which to root model comparison and selection. We based our model choices primarily upon the feature level comparisons.

## 4.3 Final evaluation

The best performing model configurations were evaluated on the held out test set. The averages and standard deviations for the scores across the three models (from each training/validation split) are presented in Table 3. For the feature measures the Spearman correlation is also given. Radar plots, showing the models' performance for each features are displayed in Fig 8,

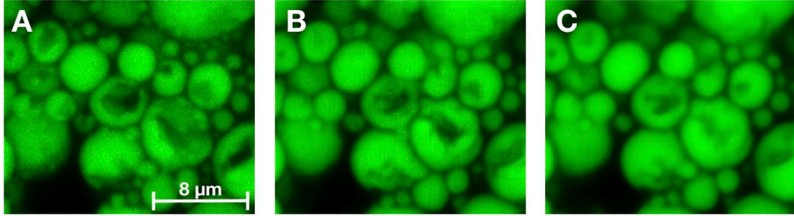

**Fig 7. Comparison of lipid defect reconstructions between the models.** (A) Ground truth fluorescence image, (B) Output from the model trained with $\mathcal{L}_1 + \mathcal{L}_{grad} + \mathcal{L}_{adv}$, (C) Output from the model trained with $\mathcal{L}_1$. In (B) there is a better delineation of the lipids and an improved reconstruction of the lipid defects/dents than in (C). Images are displayed with the same dynamic range for visualization purpose.

**Table 3. Evaluation of the models for the three channels on the held out test set, where ρ denotes the Spearman correlation averaged across the results of the three models (from each training/validation split).**

| Channel | MAE | SSIM | $CP_{Morphology}$ | | $CP_{Intensity}$ | | $CP_{Count}$ | | $CP_{Tot}$ | |
|---|---|---|---|---|---|---|---|---|---|---|
| | Score | Score | Score | ρ | Score | ρ | Score | ρ | Score | ρ |
| Nuclei | 0.47 ± 0.016 | 0.98 ± 0.001 | 0.04 ± 0.007 | 0.91 | 0.16 ± 0.070 | 0.58 | 0.09 ± 0.012 | 0.82 | 0.10 ± 0.010 | 0.77 |
| Lipid droplets | 0.14 ± 0.006 | 0.99 ± 0.001 | 0.16 ± 0.005 | 0.65 | 0.29 ± 0.010 | 0.57 | 0.210 ± 0.02 | 0.47 | 0.19 ± 0.001 | 0.52 |
| Cytoplasm | 0.18 ± 0.002 | 0.99 ± 0.000 | 0.09 ± 0.009 | 0.78 | 0.13 ± 0.022 | 0.68 | 0.15 ± 0.013 | 0.83 | 0.11 ± 0.012 | 0.59 |

where the mean values for the ground truth and the generated images are displayed along with their correlations. Note that the mean granularity for the lipid droplets is the mean of the granularity measures across the different scales.

A qualitative comparison of the results on the test data can be seen in Fig 9 which shows that the proposed models generate the fluorescent stains accurately. Fig 9A and 9D shows the generated and ground truth for the nuclei. As can be seen in the maximum projection of the bright-field z-stack (Fig 9G) generating the nuclei from the bright-field images poses significant challenges—nuclei not surrounded by lipid droplets are barely visible. However, the proposed method manages to find these nuclei well (see for instance right most nuclei in Fig 9A.1). The model also faithfully captures the shape of the nuclei, which is supported by the results in Table 2 and Fig 6, where the LWM improves significantly over the baseline model on the shape features. However, difficulties in predicting the nuclei can be seen also in Fig 9A.1 where the model misses a nucleus in a location where one should be present. Hence, although this problem of missing nuclei is reduced when using the LWM, as opposed to the base model, it has not been entirely eliminated. Occasionally, also the opposite problem occurs, where the model predicted a nucleus in a location where no nucleus was present. These errors tended to occur in small gaps in densely clustered lipid regions where nuclei are often located.

Fig 9B and 9E show the generated and ground truth images for the lipid droplets. Since the droplets are quite visible in the bright-field (see for instance Fig 9G.2) this reconstruction task is significantly simpler. The main difficulties lies in reconstructing the droplets internal structures, such as the small defects. From Fig 7 and Table 2 the reconstruction of these defects improved significantly with the gradient based loss and the adversarial training.

The results for the cytoplasmic stain can be seen in Fig 9C which show that the model manages to predict the cytoplasmic stain with high fidelity. Fig 9F.1 and 9F.2 contain some examples of intensity variations that the model does not manage to recreate. This is not surprising as there is very little information about these variations in the bright-field data. As such large intensity fluctuations are rather sparse in the cytoplasm images, modelling approaches that try to accommodate for them, such as those including gradient and adversarial loss, may lead to hiccups where the models try to recreate such variations where they should not be.

Quantitatively we see a good correlation between features extracted from the ground truth images and the reconstructed images (Table 3). Looking at the individual feature scores for the reconstructed nuclei (Fig 8) the morphological and count features line up well with those extracted from the ground truth. The main difference being in the intensity features where the intensity from the generated images is (on average) lower than that from the ground truth images. This relates back to the problem of recreating the internal structures of the nuclei, where the generated nuclear objects tended to have a more uniform intensity.

From Fig 8, for lipid droplets, the count features from the generated images were lower an average. Although adding gradient loss and adversarial training improved the model for this channel, it still missed some of the defective lipids, which consequently resulted in a lower

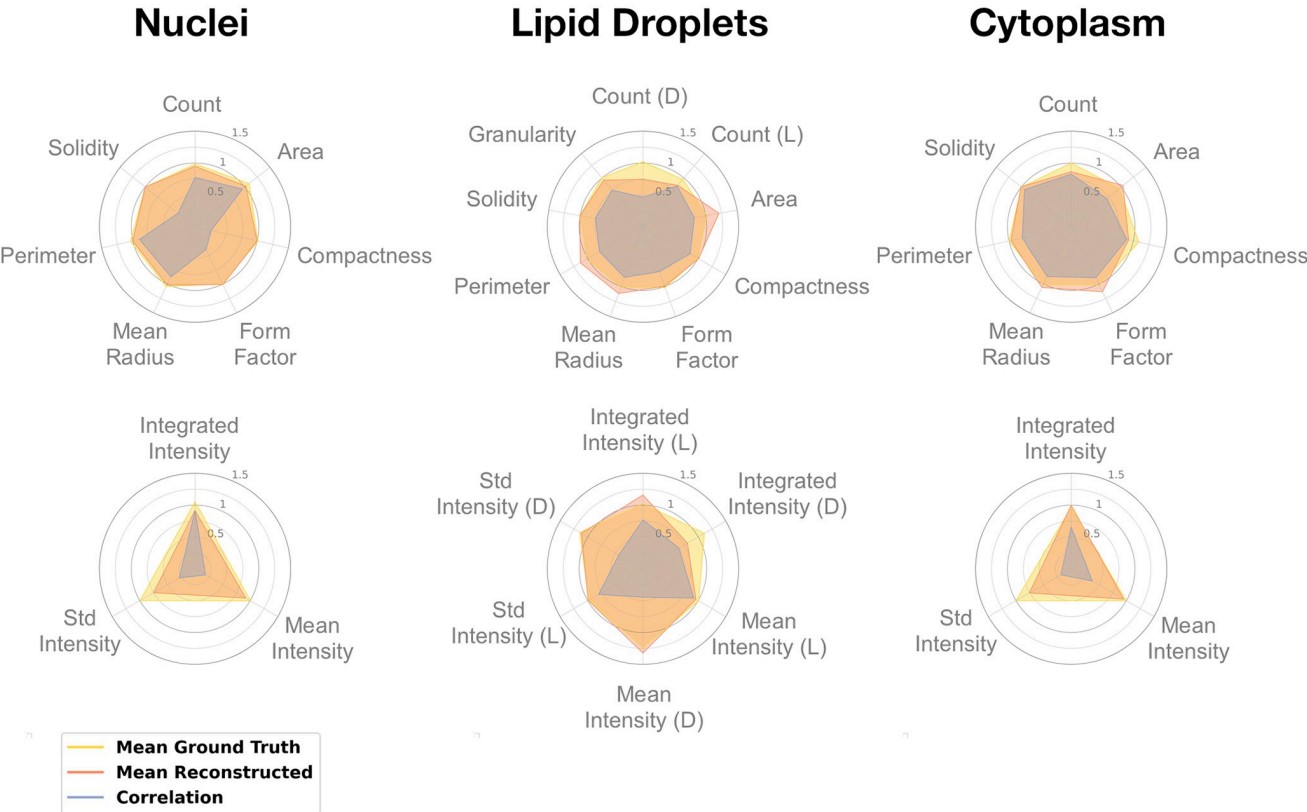

**Fig 8. Individual feature scores for the models evaluated on the test set.** The plots shows the mean of each feature across the images and display the results for both the generated and ground truth fluorescence images. The plots also include the Spearman correlation for each feature. Top row represents morphology and count features and bottom row shows intensity features. For the lipids: (D) lipids with defects; and (L) all lipid droplets.

integrated intensity. The area features based on the generated images were higher on average. This is probably due to the fact that smaller lipids were often missed in the generated images (see Fig 7). Apart from these examples the model performs almost perfectly in the remaining feature categories.

Since the cytoplasm segmentation in the CellProfiler pipeline is bound to both the nucleus and the lipid droplets the scores for this channel are effected by the performance on these other two channels. This effect can be seen for instance in Fig 8, where the compactness score for the generated images is lower than for the ground truth, i.e. it predicts more compact objects with less irregularities and holes.

## 5 Conclusion

In this work we carefully considered the challenges posed by each imaging channel to tailor our solutions, whilst also fine-tuning the model selection based on the ablation analysis. The most tailored solution was for the nuclear channel. In many cases the nuclei have only a ghost-like presence in the bright-field images (see Fig 9). We therefore chose to apply the LUPI paradigm to guide the model for this channel, using segmentation masks as privileged information to guide for the predictions. This guidance led to significantly improved results which is in line with related work using LUPI for image related tasks [22–26].

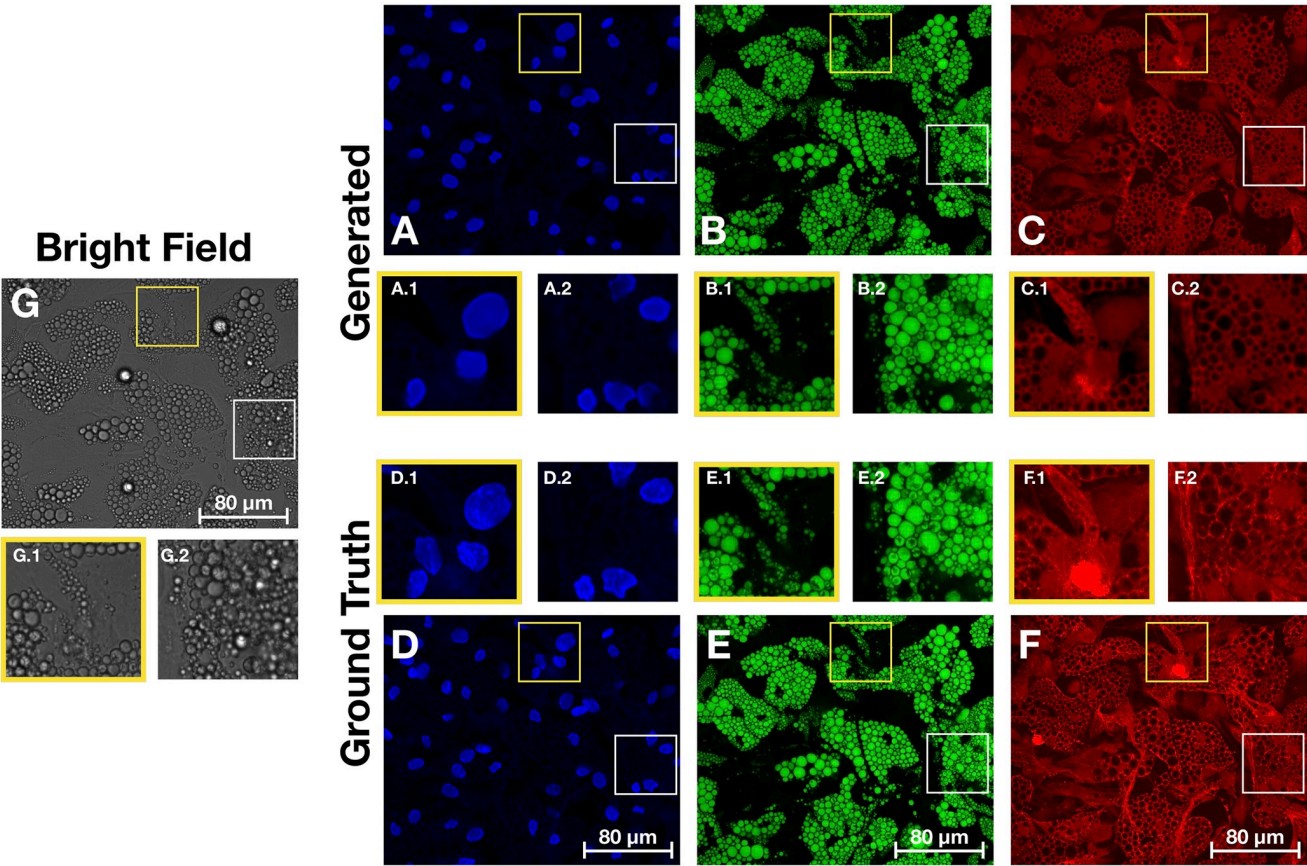

**Fig 9. Comparison of images generated from the bright-field z-stack and the ground truth fluorescence images. A-C.** Generated images for the nuclei, lipid droplets and cytoplasm, with zoomed in regions showing some well reconstructed areas alongside some more problematic locations. **D-F** Ground truth images with corresponding zoomed in regions. **G** Maximum projection of the bright-field z-stack with corresponding zoomed in regions. Images are displayed with the same dynamic range per channel for visualization purpose.

For the lipids channel we found that adding gradient and especially adversarial loss gave significant improvements in delineating the lipids and in reconstructing the defective parts within them (Figs 7 and 9). Defective lipids become caved in, with intensity depressions, and are often clustered together inside the same cell—an active cell. Locating and quantifying these defects is an important ingredient of the nanomedicine drug discovery process, hence faithfully reconstructing these intensity depressions with our models is crucial.

Although we gave attention to how well our models performed at the image-level, our main focus was on creating models that were most faithful at the extracted feature-level. Indeed, in several cases the models that gained the highest score at the image level, based on the MAE, were not those that performed best at the feature-level. As downstream analysis from fluorescence images is based on the derived features, including the design of effective drug therapies, we believe that this focus is of higher benefit when applying deep learning to the task of creating virtually stained images.

As presented in this paper, generating the nuclear, cytoplasmic and lipid droplet images directly from bright-field images liberates the fluorescence channels so that, if some perturbation of the cells can be tolerated, they can be used to visualize other aspects of cell physiology (proteins, lipid subsets, molecular reporters etc.). In the context of nanoparticles and vaccines,

for example, this enables additional measurements of particle uptake, cargo delivery and functional measurements [39]. Virtual staining also delivers results at a fraction of the cost and time required for traditional fluorescence imaging [13]. As our trained models are lightweight, and generate the virtually stained images relatively quickly, they could potentially be directly integrated into the image acquisition software connected to the microscopes to essentially enable "augmented microscopy" [40].

## Acknowledgments

A very special thank you to Alan Sabirsh from AstraZeneca, who created the adipocyte dataset and the CellProfiler pipeline, and with whom we had many enlightening discussions. And a sincere thank you to Carolina Wählby, Ola Spjuth, Andreas Hellander and Ida-Maria Sintorn from Uppsala University. Finally, thanks to AstraZeneca and AI Sweden for organizing the *Adipocyte Cell Imaging* challenge, which provided the springboard for the work in this manuscript.

## Author Contributions

**Conceptualization:** Håkan Wieslander, Ankit Gupta, Philip John Harrison.

**Formal analysis:** Håkan Wieslander, Ankit Gupta, Philip John Harrison.

**Investigation:** Håkan Wieslander, Ankit Gupta, Ebba Bergman, Erik Hallström, Philip John Harrison.

**Methodology:** Håkan Wieslander, Ankit Gupta, Philip John Harrison.

**Project administration:** Philip John Harrison.

**Resources:** Håkan Wieslander, Ankit Gupta, Ebba Bergman, Erik Hallström, Philip John Harrison.

**Software:** Håkan Wieslander, Ankit Gupta, Philip John Harrison.

**Supervision:** Philip John Harrison.

**Validation:** Håkan Wieslander, Ankit Gupta, Ebba Bergman, Erik Hallström, Philip John Harrison.

**Visualization:** Håkan Wieslander, Ankit Gupta, Philip John Harrison.

**Writing – original draft:** Håkan Wieslander, Philip John Harrison.

**Writing – review & editing:** Håkan Wieslander, Ankit Gupta, Ebba Bergman, Erik Hallström, Philip John Harrison.

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
