## [Decision Letter · Decision Letter 0]

30 Jun 2021

PONE-D-21-15339

Learning to see colours: generating biologically relevant fluorescent labels from bright-field images

PLOS ONE

Dear Dr. Harrison,

Thank you for submitting your manuscript to PLOS ONE. After careful consideration, we feel that it has merit but does not fully meet PLOS ONE’s publication criteria as it currently stands. Therefore, we invite you to submit a revised version of the manuscript that addresses the points raised during the review process.

We look forward to receiving your revised manuscript.

Kind regards,

Chi-Hua Chen, Ph.D.

Academic Editor

PLOS ONE

Journal Requirements:

2. We note that Figure(s) 6, 8, and 9 in your submission contain copyrighted images. All PLOS content is published under the Creative Commons Attribution License (CC BY 4.0), which means that the manuscript, images, and Supporting Information files will be freely available online, and any third party is permitted to access, download, copy, distribute, and use these materials in any way, even commercially, with proper attribution. For more information, see our copyright guidelines: http://journals.plos.org/plosone/s/licenses-and-copyright.

a. You may seek permission from the original copyright holder of Figure(s) 6, 8, and 9 to publish the content specifically under the CC BY 4.0 license. 

Reviewers' comments:

Reviewer's Responses to Questions

**Comments to the Author**

1. Is the manuscript technically sound, and do the data support the conclusions?

Reviewer #1: Partly

2. Has the statistical analysis been performed appropriately and rigorously? 

Reviewer #1: No

3. Have the authors made all data underlying the findings in their manuscript fully available?

Reviewer #1: No

4. Is the manuscript presented in an intelligible fashion and written in standard English?

Reviewer #1: Yes

5. Review Comments to the Author

Reviewer #1: PONE-D-21-15339

In this manuscript, the authors apply the established approach of label-free microscopy to an adipocyte dataset with two . They modify previous deep-learning approaches using a variation on deep learning called Learning Using Privileged Information. They also use a feature-based approach to evaluate generated images, rather than the pixel-based approaches used in previous label-free work. There are a three significant issues with this manuscript that need to be addressed. First, relevant prior work is not cited (as itemized below). Second, the authors did not compare their approach to current state-of-the-art methods Without these comparisons it is unclear whether or not the manuscript contributes an advance to the field. Third, the images used in the study are not being made publicly available as required by journal policy.

Major issues:

1. The title, abstract and introduction are largely written as if the authors invented the approach of label-free microscopy, which is not correct. All should be modified to make clear the relationship of their work to prior work clear.

2. The authors briefly mention Christiansen et al 2018 but do not mention the even more relevant papers from the Allen Institute for Cell Science (Johnson et al (2017) [https://arxiv.org/abs/1705.00092] and Ounkomol et al (2018) [https://doi.org/10.1038/s41592-018-0111-2]. They do not compare their approach to these state-of-the-art methods, which could been done either by applying the open source Allen Institute for Cell Science software to their images, or by applying their method to the publicly available images used by Ounkomol et al.

3. The idea of using features to evaluate synthetic images has been described previously by Zhao & Murphy (2007) [https://doi.org/10.1002/cyto.a.20487].

4. While using features is appropriate, the authors should also calculate pixel-wise correlations in order to be able to compare their results to the previous work.

5. In order to enable others to reproduce and possibly improve upon the work and to comply with PLoS ONE policy, all images used in the study should be made publicly available. This reviewer strongly urges that the manuscript not be published without making the images freely available.

6. PLOS authors have the option to publish the peer review history of their article (what does this mean?). If published, this will include your full peer review and any attached files.

Reviewer #1: No

---

## [Author Response · Author response to Decision Letter 0]

24 Jul 2021

Dear reviewer,

Thank you for carefully going through our manuscript. Please find our responses to your comments below. In the revised manuscript we have made updates in line with your comments.

Best regards, Philip J Harrison

PONE-D-21-15339

Learning to see colours: generating biologically relevant fluorescent labels from bright-field images

 Review Comments to the Author

Reviewer #1: PONE-D-21-15339

In this manuscript, the authors apply the established approach of label-free microscopy to an adipocyte dataset with two . They modify previous deep-learning approaches using a variation on deep learning called Learning Using Privileged Information. They also use a feature-based approach to evaluate generated images, rather than the pixel-based approaches used in previous label-free work. There are a three significant issues with this manuscript that need to be addressed. First, relevant prior work is not cited (as itemized below). Second, the authors did not compare their approach to current state-of-the-art methods Without these comparisons it is unclear whether or not the manuscript contributes an advance to the field. Third, the images used in the study are not being made publicly available as required by journal policy.

Major issues:

1. The title, abstract and introduction are largely written as if the authors invented the approach of label-free microscopy, which is not correct. All should be modified to make clear the relationship of their work to prior work clear.

We did not intent to make it sound like we had invented label-free microscopy. We have added text in the abstract and references to additional articles (see response to your next comment) to deal with this concern.

2. The authors briefly mention Christiansen et al 2018 but do not mention the even more relevant papers from the Allen Institute for Cell Science (Johnson et al (2017) [https://arxiv.org/abs/1705.00092] and Ounkomol et al (2018) [https://doi.org/10.1038/s41592-018-0111-2]. They do not compare their approach to these state-of-the-art methods, which could been done either by applying the open source Allen Institute for Cell Science software to their images, or by applying their method to the publicly available images used by Ounkomol et al.

We agree that these two papers should have been cited in our manuscript. The Johnson et al paper we had not previously come across. Thank you for pointing out this paper to us, their method was interesting. We have added references to these two paper (see lines 66-72).

Concerning comparing our method to other state-of-the-art methods: As we tailored our methods to each imaging channel and our specific adipocyte image data, focusing on the derived features, and were not trying to devise a general method that would work for all virtual staining, we believe it would be difficult to devise a fair comparison. We can not directly apply these alternative methods directly to our dataset without making modifications to them. For instance, the Allen Institute for Cell Science software was developed for 3D reconstructions, not 2D as was our case. We have updated the text on line 102 to emphasize that we have build this method specifically for our adipocyte cell image data. 

It is also the case that our method was actually compared against 7 other state-of-the-art methods in the competition organised by AstraZeneca and AI Sweden that we won. The 7 other teams used a variety of different and current approaches for the virtual staining. AstraZeneca and AISweden are in the process of writing a paper about the competition and the modelling solutions used by the competing teams. Unfortunately, this paper has not yet been completed nor published.

3. The idea of using features to evaluate synthetic images has been described previously by Zhao & Murphy (2007) [https://doi.org/10.1002/cyto.a.20487].

Thank you for bringing this paper to our attention, we have included a reference to it on lines 100-101.

4. While using features is appropriate, the authors should also calculate pixel-wise correlations in order to be able to compare their results to the previous work.

See previous response concerning comparing to other methods. We do however still report our pixel-level performance (MAEs) in Tables 2 and 3.

5. In order to enable others to reproduce and possibly improve upon the work and to comply with PLoS ONE policy, all images used in the study should be made publicly available. This reviewer strongly urges that the manuscript not be published without making the images freely available.

We would really like to make this data available, but unfortunately there are legal restrictions that make this impossible at the current time. At present access to the data can be requested from AI Sweden, although a sample of the data is freely available at https://www.ai.se/en/node/81535/adipocyte-cell-imaging-challenge.

---

## [Decision Letter · Decision Letter 1]

13 Aug 2021

PONE-D-21-15339R1

Learning to see colours: generating biologically relevant fluorescent labels from bright-field images

PLOS ONE

Dear Dr. Harrison,

Thank you for submitting your manuscript to PLOS ONE. After careful consideration, we feel that it has merit but does not fully meet PLOS ONE’s publication criteria as it currently stands. Therefore, we invite you to submit a revised version of the manuscript that addresses the points raised during the review process.

We look forward to receiving your revised manuscript.

Kind regards,

Chi-Hua Chen, Ph.D.

Academic Editor

PLOS ONE

Journal Requirements:

Reviewers' comments:

Reviewer's Responses to Questions

**Comments to the Author**

1. If the authors have adequately addressed your comments raised in a previous round of review and you feel that this manuscript is now acceptable for publication, you may indicate that here to bypass the “Comments to the Author” section, enter your conflict of interest statement in the “Confidential to Editor” section, and submit your "Accept" recommendation.

Reviewer #1: (No Response)

2. Is the manuscript technically sound, and do the data support the conclusions?

Reviewer #1: Partly

3. Has the statistical analysis been performed appropriately and rigorously? 

Reviewer #1: Yes

4. Have the authors made all data underlying the findings in their manuscript fully available?

Reviewer #1: No

5. Is the manuscript presented in an intelligible fashion and written in standard English?

Reviewer #1: Yes

6. Review Comments to the Author

Reviewer #1: The authors have made a few welcome additions based upon my previous comments. They have decided not to compare their method to previous methods. While I understand their choice, in the absence of such a comparison the methodological significance of the work remains unknown. Therefore the work is best described as the application of an existing approach to a different cell type. I therefore strongly feel that the title is not appropriate and should be changed to something like “Development of label-free microscopy models for adipocytes”.

7. PLOS authors have the option to publish the peer review history of their article (what does this mean?). If published, this will include your full peer review and any attached files.

Reviewer #1: No

---

## [Author Response · Author response to Decision Letter 1]

23 Aug 2021

RESPONSE TO EDITOR

I am pleased to resubmit the original paper by Håkan Wieslander, Ankit Gupta, Ebba Bergman, Erik Hallström and Philip J Harrison for publication consideration in PLOS ONE. This paper was originally entitled “Learning to see colours: generating biologically relevant fluorescent labels from bright-field images”, but in response to the reviewer’s suggestion we have now changed the title to “Learning to see colours: biologically relevant virtual staining for adipocyte cell images”.

We have reviewed our reference list to ensure that it is complete and correct. None of the cited papers have been retracted.

We hope you find our revised manuscript suitable for publication in PLOS ONE.

RESPONSE TO REVIEWER

We agree, in light of your comments, and those you made on the first draft of our manuscript that a less general title would be better. We have therefore changed the title of our manuscript to “Learning to see colours: biologically relevant virtual staining for adipocyte cell images”.

---

## [Decision Letter · Decision Letter 2]

30 Sep 2021

Learning to see colours: biologically relevant virtual staining for adipocyte cell images

PONE-D-21-15339R2

Dear Dr. Harrison,

We’re pleased to inform you that your manuscript has been judged scientifically suitable for publication and will be formally accepted for publication once it meets all outstanding technical requirements.

Kind regards,

Chi-Hua Chen, Ph.D.

Academic Editor

PLOS ONE

Additional Editor Comments (optional):

Reviewers' comments:

Reviewer's Responses to Questions

**Comments to the Author**

1. If the authors have adequately addressed your comments raised in a previous round of review and you feel that this manuscript is now acceptable for publication, you may indicate that here to bypass the “Comments to the Author” section, enter your conflict of interest statement in the “Confidential to Editor” section, and submit your "Accept" recommendation.

Reviewer #1: All comments have been addressed

2. Is the manuscript technically sound, and do the data support the conclusions?

Reviewer #1: Yes

3. Has the statistical analysis been performed appropriately and rigorously? 

Reviewer #1: Yes

4. Have the authors made all data underlying the findings in their manuscript fully available?

Reviewer #1: No

5. Is the manuscript presented in an intelligible fashion and written in standard English?

Reviewer #1: Yes

6. Review Comments to the Author

Reviewer #1: (No Response)

7. PLOS authors have the option to publish the peer review history of their article (what does this mean?). If published, this will include your full peer review and any attached files.

Reviewer #1: No

---

## [Editor Report · Acceptance letter]

7 Oct 2021

PONE-D-21-15339R2 

Learning to see colours: biologically relevant virtual staining for adipocyte cell images 

Dear Dr. Harrison:

I'm pleased to inform you that your manuscript has been deemed suitable for publication in PLOS ONE. Congratulations! Your manuscript is now with our production department. 

Kind regards, 

on behalf of

Professor Chi-Hua Chen 

Academic Editor

PLOS ONE